# Expression of Hairpin-Enriched Mitochondrial DNA in Two Hairworm Species (Nematomorpha)

**DOI:** 10.3390/ijms241411411

**Published:** 2023-07-13

**Authors:** Olga V. Nikolaeva, Aleksandra M. Beregova, Boris D. Efeykin, Tatiana S. Miroliubova, Andrey Yu. Zhuravlev, Andrey Yu. Ivantsov, Kirill V. Mikhailov, Sergei E. Spiridonov, Vladimir V. Aleoshin

**Affiliations:** 1Belozersky Institute of Physicochemical Biology, Lomonosov Moscow State University, Leninskie Gory Str., 1, Bld. 40, Moscow 119991, Russia; 2Faculty of Bioengineering and Bioinformatics, Lomonosov Moscow State University, Leninskie Gory Str., 1, Bld. 73, Moscow 119991, Russia; 3Severtsov Institute of Ecology and Evolution, Russian Academy of Sciences, Leninskiy Ave., 33, Moscow 119071, Russias_e_spiridonov@rambler.ru (S.E.S.); 4Institute for Information Transmission Problems (Kharkevich Institute), Russian Academy of Sciences, Bolshoy Karetny Per., 19, Bld. 1, Moscow 127051, Russia; 5Faculty of Biology, Lomonosov Moscow State University, Leninskie Gory Str., 1, Bld. 12, Moscow 119991, Russia; 6Borissiak Palaeontological Institute, Russian Academy of Sciences, Profsoyuznaya Str., 123, Moscow 117647, Russia

**Keywords:** Nematomorpha, mitochondrial genome, RNA-seq, transcript modifications, polyadenylation, phylogeny

## Abstract

Nematomorpha (hairworms) is a phylum of parasitic ecdysozoans, best known for infecting arthropods and guiding their hosts toward water, where the parasite can complete its life cycle. Over 350 species of nematomorphs have been described, yet molecular data for the group remain scarce. The few available mitochondrial genomes of nematomorphs are enriched with long inverted repeats, which are embedded in the coding sequences of their genes—a remarkably unusual feature exclusive to this phylum. Here, we obtain and annotate the repeats in the mitochondrial genome of another nematomorph species—*Parachordodes pustulosus*. Using genomic and transcriptomic libraries, we investigate the impact of inverted repeats on the read coverage of the mitochondrial genome. Pronounced drops in the read coverage coincide with regions containing long inverted repeats, denoting the ‘blind spots’ of short-fragment sequencing libraries. Phylogenetic inference with the novel data reveals multiple disagreements between the traditional system of Nematomorpha and molecular data, rendering several genera paraphyletic, including *Parachordodes*.

## 1. Introduction

Nematomorpha or horsehair worms can usually only be encountered during their mature life stage, as aquatic worms resembling nematodes. Most species range in length from 5 to 10 cm long, reaching 200 cm, and from 0.1 to 0.2 cm in diameter. However, these worms are only a transient reproductive non-feeding stage of the nematomorph life cycle, most of their life is spent feeding and developing as larvae and juveniles in the body cavity of their host [1]. Although sporadic human and canine cases of infection by horsehair worms have been reported (e.g., [2,3]), the group predominantly parasitizes arthropods. Currently, the phylum Nematomorpha comprises two classes: marine Nectonematoida (one genus, five species) and freshwater Gordioidea (20 genera, more than 350 species) [4]. A common tool for assisting species delimitation is mtDNA, specifically the *cox1* gene—a principal marker used in DNA barcoding [5]. Species delimitation that relies only on morphological features proves to be difficult for nematomorphs, and due to the lack of efficient cultivation techniques and the infrequency of their occurrence, molecular data for the group is limited [6,7,8,9]. Additionally, the enrichment of their mtDNA by long inverted repeats presents difficulties during amplification and sequencing [10].

The genus *Parachordodes* differs from other freshwater nematomorphs in a combination of features. Their posterior end is prominently divided into two lobes in males, and they possess two types of areoles: micrareoles and megareoles with a central pore; the subventral rows of bristles do not form cuticular ridges (adhesive warts) in the pre-cloacal region; one or more spine rows are located in the intra-areolar furrows [11]. However, some *Parachordodes* species do not have the two types of areoles, and the genus might require revision [12,13]. Nominal representatives of the genus are widespread in Asia and Europe including remote islands in the northern Atlantic Ocean (Iceland and the Faroe Islands) [6,12]; they are less common in North America and New Zealand [14,15], and the affinities of African species are not yet established. The delimitation of species is based on the areole form and the cuticle structures of the posterior end cuticle structures in males, although the variability of these features is not fully known, due to limited sampling. The principal hosts for these species are darkling beetles (genus *Blaps* and its close relatives), but there are reports of parasitism of *P. pustulosus* in ground beetles. Currently, molecular data are only available for a single species of the genus—*P. diblastus* [15].

Mitochondrial genes are transcribed as one or a few long polycistronic RNAs [16]. These polycistronic RNA precursors undergo processing, which includes RNA precursor cleavage into separate genes [16]. According to the tRNA-punctuation model, PCGs and rRNA genes are separated with one or several tRNA genes, which act as ‘punctuation marks’ for pre-RNA cleavage [17]. Not all cleavage events, however, can be explained by the tRNA-punctuation model, as not all of mRNAs are flanked with tRNAs. Unfortunately, mRNA maturation is not so well studied in non-model organisms. Gene expression regulation can take place at all stages of mRNA maturation. The peculiar prevalence of inverted repeats in the mitochondrial genomes of nematomorphs, however, presents both a challenge for their study and an interesting case of molecular evolution in itself. Here, we sequence and analyze the mtDNA of *P. pustulosus* to investigate the phylogeny of the genus and the relationships within the whole phylum. Additionally, we document the effects of inverted repeats found in the mtDNA of nematomorphs on the gene coverage in the sequencing libraries and assess the applicability of the tRNA-punctuation model for their mitochondrial transcripts.

## 2. Results and Discussion

### 2.1. Structure of Parachordodes pustulosus Mitochondrial Genome

The mitochondrial genome of *Parachordodes pustulosus* (GenBank OR003915) constitutes a single, circular DNA molecule of 14,013 bp in size, containing a typical metazoan set of 37 genes (2 rRNAs, 22 tRNAs and 13 protein-coding genes) (Figure 1; Table 1). Twenty-five of the thirty-seven genes (*cox1*, *cox2*, *trnK*, *trnD*, *atp8*, *atp6*, *cox3*, *trnG*, *nad3*, *trnA*, *trnR*, *trnN*, *trnS1*, *nad4l*, *nad4*, *trnH*, *nad5*, *trnF*, *trnQ*, *trnY*, *trnT*, *nad6*, *cytb*, *trnS2*, and *trnE*) are located on a single strand, which we designate as the (+) strand; 12 genes (*trnW*, *nad2*, *trnM*, *trnP*, *trnL1*, *rrnL*, *trnV*, *rrnS*, *trnC*, *trnI*, *nad1*, *trnL2*) are located on the other strand (−strand). The genes are positioned in a few continuous gene blocks: 12 genes from the (−) strand form three blocks—*trnW*-*nad2*-*trnM*, *trnP* and *trnL1*-*rrnL*-*trnV*-*rrnS*-*trnC*-*trnI*-*nad1*-*trnL2*. On the basis of sequence alignments, we predict five types of start codons in the protein coding genes (PCGs) (Table 2). The majority of PCGs are predicted to start with codons ATG or ATA. Unconventional start codons are specific for some genes: *atp8* is likely initiated by the isoleucine codon ATC, *nad2* and *nad5* are likely initiated by the leucine codon TTA, and *nad1* is likely initiated by the leucine codon TTG. Twelve PCGs are terminated by two stop codons (TAA and TAG) with a preference for TAA. The *nad3* gene is terminated by an incomplete codon T (position 4380), which is completed to TAA after polyadenylation of the mRNA. The polyadenylation in this place was confirmed by found transcripts with poly-A tails (Figure 2).

In total, we found 24 polyadenylation sites in both strands (Figure 1). Polyadenylation takes place after RNA precursor cleavage, so all polyadenylated ends mark potential cleavage sites. We see that almost each PCG border can be such a site (excluding pair *nad6*-*cytb*), and, if cleaved altogether, transcripts containing these PCGs will be monocistronic. Eight mRNAs are polyadenylated (at least in some proportion) immediately after the open reading frame. Four mRNAs have either polyadenylation immediately after the open reading frame, or 3’ UTR containing one polyadenylated tRNA (*cox2*-*trnK*, *nad4*-*trnH*, *nad2*-*trnW*) or two tRNAs (*nad3*-*trnA*-*trnR*) with the polyadenylation after the last one in each transcript. We also found out that many RNA-seq reads, mapped to some tRNA genes (*trnK*, *trnD*, *trnA*, *trnS1*, *trnH*, *trnY, trnS2, trnP*, *trnV*), have three additional nucleotides (CCA) after 3′ ends of genes, which do not coincide with a reference genome sequence (Appendix A). These supplementary nucleotides are followed by polyadenylation, so we propose that polyadenylation takes place after non-template adding of CCA at least in some cases.

*Gordionus alpestris* has the same gene order as *P. pustulosus* so we can compare their polyadenylation sites. We found nine polyadenylation sites in the mitochondrial genome of *G. alpestris* (Figure 1). All of them coincide with polyadenylation sites in the *P. pustulosus* mitochondrial genome. Both transcriptomes lack polyadenylation between *nad6* and *cytb*, so the mRNA containing them appears to be bicistronic. We predict that, unlike *P. pustulosus*, in *G. alpestris*, there are another two bicistronic mRNAs combining atp8 + atp6 and nad4l + nad4.

The GC content in the tRNA (23%) and rRNA (21%) genes is lower than the average GC content in the whole mitochondrial genome (27%) (Table 2); GC content in the PCGs (29%) slightly exceeds the average. The prevalence of thymine over adenine and cytosine over guanine in the (+) strand provides negative AT-skew (−0.096) and GC-skew (−0.11).

The mitochondrial genome of *P. pustulosus* contains multiple perfect inverted repeats, including repeats longer than 100 bp in length. In total, there are 55 hairpins longer than 10 bp in the mtDNA of *P. pustulosus*. Some of the repeats overlap each other, thus forming 29 distinct hairpin regions in the mtDNA of *P. pustulosus*. Most of the hairpin regions are located in the PCGs: *atp6*, *cox3*, *nad1*, *nad2*, *nad4*, *nad4l*, *nad5*, *nad6* (Table 1). The PCGs *cox1*, *cox2*, *atp8*, *nad3* are free of inverted repeats, as are the majority of tRNA genes.

The non-coding regions are 624 bp in total and consist of 22 intergenic segments, ranging from 1 to 175 bp and including two major non-coding regions of more than 100 bp. The largest non-coding region (between *trnL2* and *cox1*) separates the gene blocks with opposite transcription directions and potentially contains a pair of divergent promotors (Figure 3). Such regions have been previously found in the other five nematomorph mitochondrial genomes [10,18]; in each of them, they are the longest non-coding regions as well. These regions show no tandem repeats, no significant bias in their nucleotide composition compared to the whole genomes, no obvious similarity in sequences in hairworm species and they vary in length from 96 bp in *Gordius* sp. to more than 659 bp in *Gordionus_wolterstorffii*. In *G. alpestris* and *G. montsenyensis*, these regions are very similar and possess typical control region features: two imperfect direct repeats of 10 bp and some microsatellite-like structures (for example, CTT). According to these data, we consider this region to be a putative control region.

### 2.2. Mitochondrial Gene Order in Gordioida

The arrangement of protein coding and rRNA genes in the mitochondrial genome of *Parachordodes pustulosus* is identical to that seen in three other nematomorph species—*Gordionus alpestris*, *G. wolterstorffii*, and *Gordius* sp. (Figure 4). The gene order in *Chordodes* sp. differs in one transposition between the *nad6* and *nad2* genes.

The gene orders in mitochondrial genomes of nematomorphs largely retain bilaterian conservative gene blocks [19]. All available mitochondrial genomes of nematomorphs retain conservative Block-2 (*nad4l-nad4-nad5*) and most of Block-1 (*nad2-cox1-cox2-atp8-atp6-cox3-nad3*). The *nad2* gene of Block-1 is separated from other Block-1 genes and is located on the opposite strand, which indicates its transposition and inversion relative to the ancestral gene order. The nematomorphs also retain conservative Block-4 (*nad6-cytb*), with the exception of *Chordodes* sp., in which the block is divided by the transposition of the *nad6* and *nad2* genes; thus, the gene order in *Chordodes* sp. is more apomorphic. Instead of the ancestral order *nad1-rrnL-rrnS* in Block-3, all nematomorphs have the order *rrnL-rrnS-nad1*, which indicates the transposition of *nad1* within Block-3. Based on the retained conservative blocks, it is possible to reconstruct the putative gene order in Nematomorpha before the last common ancestor of Gordioida (Figure 4C). Further elucidation of the gene order evolution for Nematomorpha might be possible after obtaining data on the gene order of Nectonematoida—a deeply diverging branch of horsehair worms and sister to the Gordioida.

### 2.3. Mapping of Sequencing Libraries to the mtDNA of Parachordodes pustulosus

#### 2.3.1. Inverted Repeats and Read Coverage

The mapping of sequencing reads from a genomic DNA library shows that long inverted repeat regions are associated with noticeable drops in read depth (Figure 5A). However, smaller inverted repeats do not affect the DNA-seq coverage in a similar degree. Neither hairpin regions nor drops in DNA-seq read depth are specifically associated with gene boundaries (Figure 5A,C).

In contrast to the DNA library, the read depth of the RNA-seq library varies significantly for parts of the mitogenome. High RNA-seq coverage is observed for genes without inverted repeats (*cox1*, *cox2*, *nad3*), or genes with 1–2 repeats (*atp6*, *cox3*, *cytb*), or in rRNA genes (Figure 5A,C); in other mitogenome loci, the read depth is lower. Twenty-five hairpin regions of the total twenty-nine are located in PCGs, and all of them are associated with a decrease in RNA-seq read coverage (Figure 5A,C). As a result, the RNA-seq coverage for some genes is partial (especially for complex I genes, *nad4*, *nad5*, *nad2*, *nad6*, *nad1*). Only few of these loci are associated with drops in DNA-seq coverage (including repeats in *cox3*, *cytb*, and *rrnL)*. The differences in read coverage drops between the sequencing libraries could be a consequence of hairpins affecting not only the amplification and sequencing stages, but also the stage of reverse transcription of the RNA-seq library. RNA can easily form hairpins because it is mostly present in single-stranded form and thus blocks access for the reverse transcriptase. However besides in vitro artefacts the equal amount of genes DNA and unequal amount of genes transcripts in vivo may also influence on differences between DNA-seq and RNA-seq libraries. Moreover, 3′ and 5′ ends of transcripts are not randomly distributed which effects on 3′ and 5′ ends presence in RNA-seq output. 

#### 2.3.2. Gene Borders and RNA-Seq Coverage

A whole mitochondrial genome, including gene borders, is covered by RNA-seq reads which points to the presence of long RNA precursors. However, we see unevenness in the coverage of distinct mitogenome regions, which can be explained by several factors. Firstly, read depth toward the 3′-end of genes is usually relatively high (Figure 5A,G). This is conditioned by the cDNA library preparation technique, in which we used polyadenylated transcripts as a matrix for first chain synthesis. Secondly, the ends of DNA and cDNA fragments are expected to be under-represented due to the properties of the Illumina platform. Finally, the inverted repeat-containing parts are also associated with significant declines in RNA-seq coverage. We suggest that inverted repeats affect both the reverse transcription stage and the sequencing because of hairpin formation in vitro. For example, there is a gradual decrease in coverage of cDNA reads towards the 5′ end in the *nad6-cytb* pair in *G. alpestris*, and a sharp drop in the region of inverted repeats, which locates 40 nucleotides upstream of the *nad6* stop codon. Approximately the same coverage of cDNA reads of the 3′ end of nad6 and the adjacent downstream *cytb* region indicates the absence of cleavage between them and their fusion into a single mature bicistronic mRNA similar to those in many other species. In *P. pustulosus*, both *nad6* and *cytb* have low cDNA read coverage, and the drop in coverage between them is not clearly visible.

Longer transcripts end at the 3ʹ ends with polyadenylated tRNA sequences; shorter transcripts end in poly-A immediately after the stop codon or after a small 3ʹ UTR (e.g., 6 nucleotides in the *cox2* transcript). Transcripts of the same gene with or without tRNA at the 3ʹ end are consistent with the “reverse cleavage” version of the “tRNA punctuation” model [20,21]. The reverse gradient in cDNA read coverage in a tRNA genes block is another sign of “reverse cleavage” as in the *nad3-trnA-trnR-trnN-trnS1* block (Figure 5G). A forward gradient with increasing cDNA coverage appears to be in the *trnD-atp8* block (Figure 5A), and this may be an example of the implementation of a “forward cleavage” model [21,22] as far as we can see using Illumina’s short reads instead of long PacBio reads and full-length transcripts.

Severe changes in RNA read depth are also observed between PCGs even if there are no tRNA genes or inverted repeats in these regions. Three of these regions (*cox1*-*cox2*, *atp6*-*cox3*, and *nad4*-*nad4l*) possess polyadenylation sites after each open reading frame, so we suggest that mature mRNA coding by these genes ais monocistronic (Figure 5A,B,G,H). These transcripts are cleaved during maturation by a mechanism other than tRNA punctuation.

#### 2.3.3. Comparison of RNA-Seq of *Gordionus alpestris* and *Parachordodes pustulosus*

*P. pustulosus* and *G. alpestris* have 55 and 60 inverted repeats longer than 10 bp, respectively. Only one inverted repeat in *G. alpestris* is encountered in a non-coding region. Seven repeat-containing regions are located in a similar position in the two mitogenomes (in genes *atp6*, *nad1*, *nad2*, *nad4*, *nad6, cytb*, and *rrnL*). In both species, the repeats in *cytb* and *rrnL* are associated only with a decline in RNA-seq coverage, whereas other repeats are longer and are associated with drops in both RNA-seq and DNA-seq read depths.

Our methodology of inverted repeat research is based on in silico transcript refinement and may not be free from sequencing bias, which appears at both the reverse transcription and PCR stages. Direct RNA sequencing without a cDNA-synthesis stage, especially with long reads, will probably shed light on real nematomorph mitochondrial transcripts existing in vivo. In that case, we would be able to predict the influence of inverted repeats on mtDNA expression in nematomorphs more clearly.

### 2.4. Phylogeny

Maximum likelihood (ML) and Bayesian inference (BI) with the concatenated mitochondrial PCGs of five nematomorph species resulted in the same tree topology with high posterior probabilities and bootstrap supports (Figure 6). Nematomorphs are split up into two sister groups which approximate two freshwater nematomorph orders—Chordodea and Gordea. Two chordodeans, *Parachordodes pustulosus* and *Chordodes* sp., group together and emerge as a sister species to *Gordionus wolterstorffii*.

For a more comprehensive scope of nematomorph diversity, we reconstructed phylogeny using the available SSU rRNA and *cox1* gene data. In the reconstruction, two *Gordionus* species (*G. alpestris* and *G. montsenyensis*) group with *Gordius* spp. and *Gordionus wolterstorffii* groups with chordodeans, similarly to the mitochondrial protein tree. However, both the Chordodea and Gordea orders emerge as paraphyletic (Figure 7). The *Gordionus* species form three separate clades in the tree. Other genera (*Chordodes* and *Parachordodes*) are also paraphyletic. Instead of grouping with *P. diblastus, Parachordodes pustulosus* groups with two *Gordionus* species (*G. kii* and *G. chinensis*) and two *Paragordionus* species (*P. dispar* and *P. rautheri*).

### 2.5. Nematomorphs and Extinct Free-Living Early Vermiform Ecdysozoans

Despite their derived parasitoid lifestyle, nematomorphs retain certain features that allow us to compare them with the earliest free-living vermiform ecdysozoans [23], especially those with long and slender Cambrian-Silurian palaeoscolecidans. Some palaeoscolecidan species had a stiff cuticle consisting of microplates comparable by size and shape with those of adult nematomorphs [24,25]. Moreover, such plates bear a system of tiny orthogonally arranged furrows on the inner side [26] similar to that of nematomorphs (e.g., *Gordionus* (“*Parachordodes*”) *wolterstorffii* in [27]). In extant worms, such furrows house a cylindrical trellis of layers of collagen fibers wound around the body in left- and right-hand helices, which allow the animal to bend, coil, elongate and shorten during its locomotion [27].

Palaeoscolecidans are also characterized by the presence of an introvert (retractile proboscis) armed with spines and arranged in concentric rings [25,28]. Parasitoid nematomorph larvae have preserved an introvert surrounded by concentric rings of spines (e.g., [14,29]). Such larvae show an overall similarity to Cambrian vermiform larvae with introverts, such as *Shergoldana* and *Orstenoloricus*, which can represent larval palaeoscolecidans [30,31].

## 3. Materials and Methods

### 3.1. Sample Sources and Description

Four males and three females of *Parachordodes pustulosus* were collected in Mongolia, Gobi-Altay Aimak, near Bayan Gol River (46.70197° N, 96.31077° E, altitude 1899 m), 15–18 June 2022. All individuals were isolated from host *Blaps* sp. (Tenebrionidae family) following Medvedev [32]. The worms, which were leaving the darkling beetles after their falling in the water, were observed directly. Due to significant differences in parasite and host lengths and shapes, an intense twisting of the beetle in the water during this event occurred. By that time, the host was dead. Host individuals were found in a small, shallow (0.5 m in diameter, 0.2 m deep), artificial pit dug for the daily needs of an expedition in a seasonally dry wadi of the Bayan Gol River, where beetles as well as various other animals (mostly birds and insects) were concentrated for watering. Living worms were transferred immediately to a closed vessel and preserved in 96% ethanol.

### 3.2. SEM

Two fragments of body were cut off with a blade from the central part of the worm. The posterior end of male was also cut off 1.5–2 mm from the cloacal opening. These body fragments were dehydrated through a graded ethanol series and acetone and dried under a critical point in a HCP-2 Hitachi drier (Hitachi Ltd., Tokyo, Japan). Dried specimens were coated with gold in a BIO-RAD SC502 sputter coater (Bio-Rad Laboratories Inc., Hercules, CA, USA) and examined with a Mira 3TESCAN scanning electron microscope (SEM) at 10.0 kV in the Tescan Orsay Holding, a.s., Brno, Czech Republic. Additionally, a complete worm was studied using a TESCAN VEGA3 SEM under a similar regime in the Borissiak Palaeontological Institute, Russian Academy of Sciences, Moscow, Russia.

### 3.3. Taxonomy

*Parachordodes pustulosus* (Baird, 1853)

(Figure 8A–G)

**Material:** three males and three females.

**Locality:** Mongolia, Gobi-Altay Aimak, Khasagt Khairkhan Uul Range, Bayan Gol River (46.70197° N, 96.31077° E, altitude 1899 m), 15–18 June 2022.

**Host:** *Blaps* sp. (Tenebrionidae family).

**Males.** The body length is 106–192 mm, with a maximal body diameter of 0.5–0.7 mm. The body is dark brown in color, distinctly tapering toward the anterior end, with smooth transverse bulges along the body. The anterior end is trapezoidal, slightly flattened with lighter coloration around the apical part, with indistinct areoles; the posterior end is divided into two lobes. The mouth opening is terminal, surrounded inside by a ring of radial cuticular folds. The cuticle surface has two types of areoles: micrareoles and megareoles with a central pore. Megareoles are scattered throughout the body as singular entities or adjoined in groups, which appear as darker spots under the dissecting microscope. Surprisingly, darker coloration of megareoles can also be seen under SEM (Figure 8C). The megareoles are also present along the mid-ventral line (Figure 8D). Micrareoles are usually transversally elongated, 10–15 × 12–20 μm, and elliptical or polygonal in shape (Figure 8B). Transversal fissures are present on the surface of micrareoles. The intra-areolar furrows have numerous curved 2–3 μm long spines. These spines are not situated in one row but form two to three irregular rows along the furrow. Thinner and longer (4 μm) spines are in a furrow around the megareoles, with tips usually directed toward the pore (Figure 8B). The posterior end has two subventral bristle fields of curved, sometimes bifurcated, 6–10 μm long bristles (Figure 8E). The cloacal opening is encircled by similar curved 3–10 μm long spines, tapering toward the tip with a single bifurcated tip (Figure 8F). These spines are visibly longer on the anterior margin of the cloaca and are absent on the posterior part of the cloacal opening. The subventral surface of caudal lobes has short, robust, 2–3 μm-long conical spines. The ventral surface of the lobes has tiny conical spines of 0.5–1 μm diameter.

**Females.** The body length is 194–237 mm, with a maximal body diameter of 0.91–0.95 mm. The body is yellow brownish in color and is much lighter than in males. The flattened anterior end is white, forming a ‘calotte’, and is separated from the posterior yellow-brownish surface with a 0.6–0.9 long ring of dark cuticle. The posterior end is divided into two unpronounced lobes by a shallow, ventral groove of darker coloration, with the cloacal opening on the bottom of the groove. The cuticle surface has two types or areoles, but the megareoles do not form dark spots and have the same coloration as the micrareoles. The micrareoles are rounder than in males and are 10 × 10–12 × 15 μm in size. The megareoles are flatter than in males with smaller pores. Spines in the intra-areolar furrow are less numerous than in males (Figure 8A). Thinner spines encircle the megareoles with tips directed toward the pore.

**Taxonomic discussion**. The representatives of the genus *Parachordodes* Camerano, 1897 together with *Gordius* Linneus are some of the most common nematomorphs of the temperate zone. The diagnostic feature of this genus, according to [33], is the presence of two types of areoles: micrareoles and megareoles [34]. The pore of the duct opens in the central part of the megareole. The male tail is divided into two moderately elongated lobes. The cloacal opening is surrounded by spines. Two bristle fields are present on the subventral parts before and on the cloaca level. The post cloacal spines are present on the ventral and subventral surfaces of lobes. Numerous intra-areolar spines in males and less numerous intra-areolar spines in females are another diagnostic feature of *Parachordodes*. All these diagnostic characteristics were observed in the Mongolian nematomorphs. Thus, their identification as representatives of the genus Parachordodes is reasonable.

The discrimination of the species belonging to this genus is less sound, as the taxonomic value of some diagnostic features is still unknown. e.g., bristle fields nearly meet at the mid-ventral line in *P. pustulosus* but merge in *P. tolosanus* [33]. The bristle fields visibly meet at the mid-ventral line in our specimens, but still the species identification as *P. pustulosus* seems more reliable. This latter species is widely distributed through all Eurasia, whereas *P. tolosanus* was mainly reported in Europe. The width of intra-areolar furrow in our specimens looks more similar to that of *P. pustulosus*, as these are wider in *P. tolosanus*. Though the body size of nematomorphs is very variable, we can note that the sizes of the studied specimens are very close to those described for *P. pustulosus* (males with 100–150 mm long bodies and maximal diameter of 0.8 mm and females with 200–250 mm long bodies and 0.8 mm maximal diameter [33]). *P. tolosanus* specimens are usually smaller. Females of *P. tolosanus* are characterized by the presence of micrareoles only in the middle and posterior part of the body; whereas, in our specimens, megareoles are scattered along the body up to the posterior end.

### 3.4. DNA Sequencing and Assembly of Mitochondrial Genome

The tissues of ethanol-fixed specimens of *P. pustulosus* were manually released from the cuticle before DNA and RNA extraction. Total DNA extracts were obtained using a Diatom DNA Prep kit (Isogen, Moscow, Russia). A DNA sequencing library for *Parachordodes pustulosus* was prepared from the total DNA extract following the Accel-NGS WGA library preparation protocol. A paired-end library was sequenced using the Illumina NextSeq system, generating 31.7 M reads. The assembly was performed with SPAdes [35] after trimming the adapter sequences from the read data with Trimmomatic [36]. Mitochondrial contigs were detected in the assemblies using BLAST searches [37] with the standard set of metazoan mitochondrial-encoded protein sequences. The fragmented mitochondrial sequences were extracted from the assemblies and used as seeds for NOVOPlasty [38], resulting in the assembly of overlapping contigs, which were then pieced together to produce a complete mitochondrial genome. The reconstructed mitochondrial genome was screened for assembly artifacts by mapping the read data with bowtie2 [39] and inspecting the alignments visually using Tablet alignment viewer [40]. An invertebrate mitochondrial code was confirmed for *Parachordodes pustulosus* by both GenDecoder v1.6 [41] and FACIL [42]. The mitochondrial genome was annotated using MITOS web server [43]. The annotation was refined manually using alignments of protein-coding sequences in the BioEdit program [44]. The mitochondrial genome map was constructed using Circos software (http://circos.ca/) [45].

### 3.5. RNA Sequencing

Total RNA extracts from individual specimens (after removal of the cuticle) were obtained using TRIzol reagent lysis followed by purification with the RNeasy kit (QIAGEN, Germantown, MD, USA). The sequencing library was prepared from total RNA extracts using a TruSeq RNA library preparation kit and sequenced with Illumina NextSeq system. The libraries for differential expression analysis were constructed using the stranded version of the protocol without poly-A selection. We obtained 22.1 M read pairs (150 bp reads). The reads were trimmed with Trimmomatic and mapped to the mitochondrial genome assembly of *P. pustulosus* with bowtie2. The per gene read counts were calculated using the BEDTools genomecov utility [46].

Polyadenylation sites were found in the RNA-seq read map in Tablet alignment viewer. After the visualization of the *P. pustulosus* RNA-seq read map with Tablet alignment viewer, there was just one noticeable peak in coverage, which was located in the *rrnL* gene (10947:11266 region). Similarly, after read mapping and coverage visualization of the *G. alpestris* RNA-seq, there was one visible coverage peak in the *rrnL*, *trnV* and *rrnS* genes (11784:13691 region). Other areas of mitochondrial genomes have evenly low coverage. To obtain more data on whole mitochondrial genome expression, we masked (replaced with N) nucleotides in corresponding regions. After that, we repeated the RNA-seq mapping procedures. Because the coverage turned out to be zero in the masked areas, we manually set the maximum values of the RNA-seq there (8000 for *P. pustulosus*, 35,000 for *G. alpestris*).

Using the BEDTools utility, we found out how many read 5′ and 3′ ends per site there are in the DNA-seq and RNA-seq of both worms. In each site, we divided the amount of read 5′ and 3′ ends by the read depth; based on the results, we drew a chart for RNA read 5′ and 3′ ends per coverage per site distribution. However, there were not many peaks in the chart for DNA reads, so we just marked them with blue lines on the RNA read ends chart (charts on Figure 4B,E). Bowtie mapped reads on linear mitochondrial genomes, which are circular in vivo, which is why at the end and at the beginning of our fragment (the first and the last nucleotide) all reads had either their 5′ end or their 3′ end, so we noticed peaks on the charts (Figure 4B,E). To avoid mistakes, we provided a DNA permutation: we cut mtDNA at the beginning of the *rrnL* gene and repeated the mapping procedure. The peaks became smaller but did not disappear at all (Appendix A). 

To investigate inverted repeats (palindromes) in *P. pustulosus* and *G. alpestris* mtDNA, we used the palindrome utility of the EMBOSS package [47]. This program detects inverted repeats satisfying the given threshold of minimum and maximum length of palindrome, maximum gap between repeated regions and number of mismatches allowed. We set the following parameters: minimum length of palindrome was 10; maximum length of palindrome was 300; maximum gap between repeated regions was 100; number of mismatches allowed was 0. We also allowed palindromes to overlap.

We combined the data on RNA-seq, DNA-seq and inverted repeat location for both hairworms on one map. Also, we added gene maps of *P. pustulosus* and *G. alpestris* to Figure 3.

### 3.6. Nucleotide Composition Analysis

AT and GC skew were determined for the complete mitochondrial genomes (major strand) according to the formula AT-skew = (A − T)/(A + T) and GC-skew = (G − C)/(G + C) [48], where the letters stand for the absolute number of the corresponding nucleotides in the sequences. Characterization of codon usage bias was calculated using the BioEdit program.

### 3.7. Secondary Structure Prediction

The secondary structure of the longest non-coding region in the mitochondrial genome of *Parachordodes pustulosus* was predicted using RNA structure software (https://rna.urmc.rochester.edu/RNAstructureWeb/Servers/Predict1/Predict1.html) [49] with default values.

### 3.8. Phylogeny

The mitochondrial protein-coding gene sequences of *Parachordodes pustulosus* were translated using the invertebrate mitochondrial genetic code. The rDNA of *P. pustulosus* contig was detected in the SPAdes assembly using BLAST searches (Appendix A). This contig has a length of 9151 bp and contains complete rRNA genes *18S*, *5.8S*, and *28S* and two spacers—ITS1 and ITS2. The reconstructed rDNA contig was screened for assembly artifacts by mapping the read data with bowtie2 and inspecting the alignments visually using Tablet alignment viewer. Additional nematomorph mitochondrial PCGs and *18S* rRNA gene sequences were obtained from the GenBank database. Individual alignments of each mitochondrial PCG and *18S* rRNA were prepared using the MAFFT online service [50]. For tree inferences, the alignments were manually masked in BioEdit and concatenated in two sets: 12 mitochondrial PCGs (5 nematomorphs and 2 outgroup species) and *SSU rRNA* + *cox1* (36 nematomorphs and 4 outgroup species). *SSU rRNA* + *cox1* alignment was used for tree inference as a partitioned super matrix. Evolutionary models for phylogenetic inferences were selected using the IQ-TREE [51] ModelFinder approach (-m MF) for all alignments and MrBayes 3.2.6 [52] for amino acid alignments (aamodelpr = mixed). The best models according to IQ-TREE were GTR+F+R3 for *SSU rRNA*, mtZOA + Γ4 for *cox1*, and mtART + F + Γ4 for the 12 mitochondrial PCGs. The best model according to MrBayes 3.2.6 was MtREV for both amino acid alignments. Bayesian phylogenetic inferences (BI) were performed using MrBayes 3.2.6 with four runs under the GTR+Γ4+I for *SSU rRNA* partition and MtREV + Γ4 + I model for *cox1* partition, 5,000,000 generations and 50% burn-in. Average standard deviations of split frequencies were less than 0.007 on run completions. Maximum likelihood (ML) bootstrap support was estimated in 100 replicates (-b 100) using IQ-TREE under the GTR+F+R3 (for *SSU rRNA*), mtZOA+Γ4 for *cox1*, and mtART + F + Γ4 for the 12 mitochondrial PCGs based on Bayesian trees (-te). Phylogenetic trees were visualized with MEGA 6.0 [53].

## 4. Conclusions

The mitochondrial genome of *Parachordodes pustulosus* is similar to the genomes of previously described nematomorphs and contains multiple long inverted repeats in the coding regions of genes. The comparison of DNA-seq and RNA-seq read mappings revealed pronounced drops in coverage for long inverted repeat regions. Drops in RNA-seq coverage coincide with drops in DNA-seq coverage, but also occur in other regions, such as tRNA genes and PCG borders. Bayesian and ML tree inferences show paraphyly of the genus *Parachordodes*, placing two of its species in different clades with high support.

## Figures and Tables

**Figure 1 ijms-24-11411-f001:**
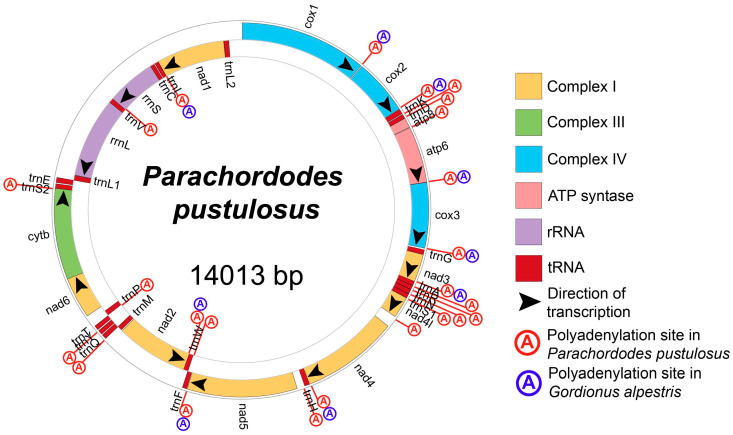
Circle map of the *Parachordodes pustulosus* mitochondrial genome.

**Figure 2 ijms-24-11411-f002:**
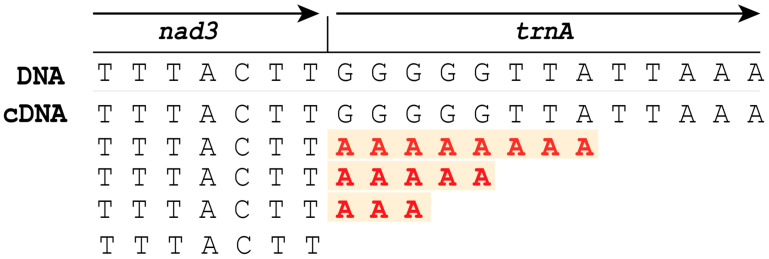
The polyadenylated transcripts aligned onto a fragment of *P. pustulosus* mitochondrial DNA. Poly-A tails are marked red.

**Figure 3 ijms-24-11411-f003:**
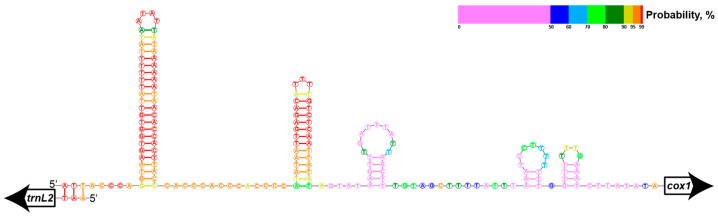
Predicted secondary structure of the longest non-coding region (a putative control region) in the mitochondrial genome of *Parachordodes pustulosus*. Probabilities of predicted secondary structure per nucleotide shown via color map.

**Figure 4 ijms-24-11411-f004:**
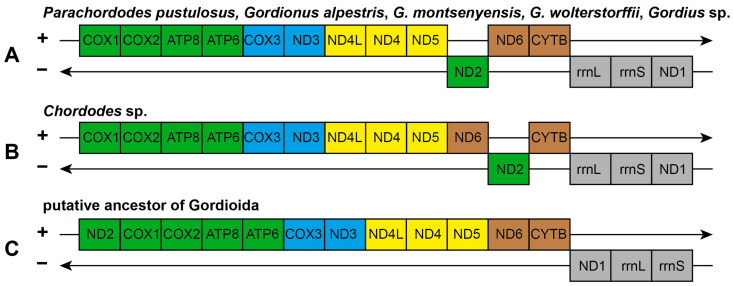
Nematomorphan mitochondrial gene orders and their relationship to bilaterian conservative gene blocks. (**A**) Gene order for the mitochondrial genomes of *Parachordodes pustulosus*, *Gordionus alpestris*, *G. wolterstorffii*, and *Gordius* sp. (**B**) Gene order for the *Chordodes* sp. mitochondrial genome. (**C**) Putative ancestral gene order of Gordioida. Colors according conserved gene blocks at [19].

**Figure 5 ijms-24-11411-f005:**
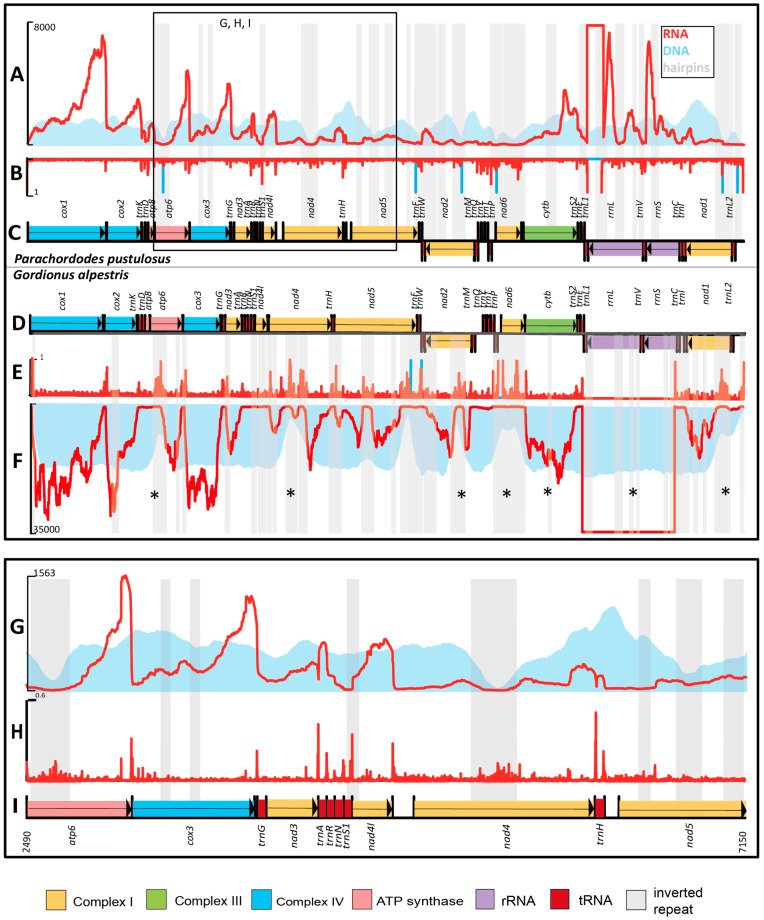
DNA-seq and RNA-seq reads mapping to the mtDNA of *Parachordodes pustulosus* (**A**–**C**) and *Gordionus alpestris* (**D**–**F**) with detailed analysis of *Parachordodes pustulosus* mtDNA region 2490:7150 (**G**–**I**). (**A**) Histograms of DNA-seq (blue) and RNA-seq (red) read counts per site in the mtDNA of *Parachordodes pustulosus*. (**B**) The ratio between 3′ + 5′ read ends of DNA-seq (blue) and RNA-seq (red) and read coverage per site in *P. pustulosus*. (**C**) *P. pustulosus* mitochondrial gene order. (**D**) *G. alpestris* mitochondrial gene order. (**E**) The ratio between 3′ + 5′ read ends of DNA-seq (blue) and RNA-seq (red) and read coverage per site in *Gordionus*. (**F**) Histograms of DNA-seq (blue) and RNA-seq (red) read counts per site in the mtDNA of *G. alpestris*. (**G**) Number of DNA-seq (blue) and RNA-seq (red) reads per site. (**H**) The ratio between 3′ + 5′ read ends of DNA-seq (blue) and RNA-seq (red) and read coverage per site. (**I**) *P. pustulosus* mitochondrial gene order in the 2490:7150 region. Inverted repeat regions are marked in light grey. Inverted repeats with a similar position in the mitochondrial genomes of *P. pustulosus* and *G. alpestris* are marked with an asterisk (*).

**Figure 6 ijms-24-11411-f006:**
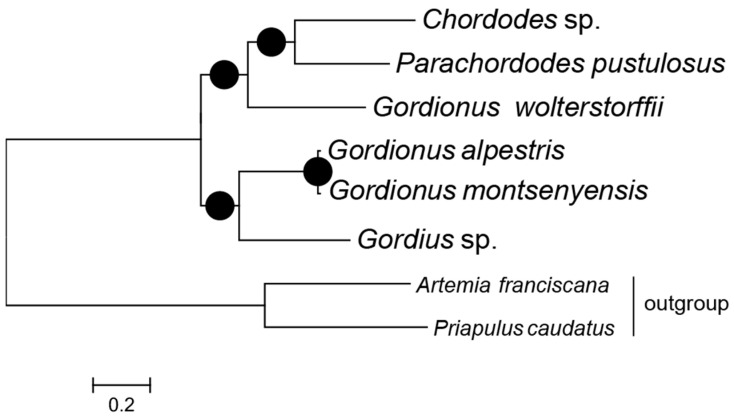
Bayesian tree of Nematomorpha based on concatenated amino acid sequences of 12 mitochondrial proteins. Bipartitions with support BI posterior probabilities/ML bootstrap support > 0.95/75 are labeled with a dot (●).

**Figure 7 ijms-24-11411-f007:**
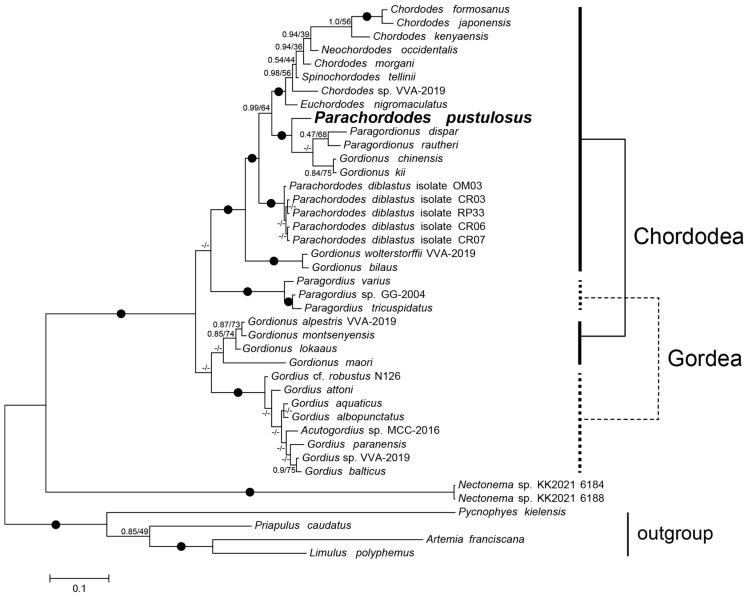
Bayesian tree of Nematomorpha based on the concatenated nucleotide (*SSU rRNA* nuclear ribosomal gene) and amino acid (mitochondrial *cox1* gene) sequences. Nodes mismatched in BI and ML topologies are unlabeled (-/-). Otherwise, labels contain BI posterior probabilities (left) and ML bootstrap support (right). Bipartitions with support > 0.95/75 are labeled with a dot (●).

**Figure 8 ijms-24-11411-f008:**
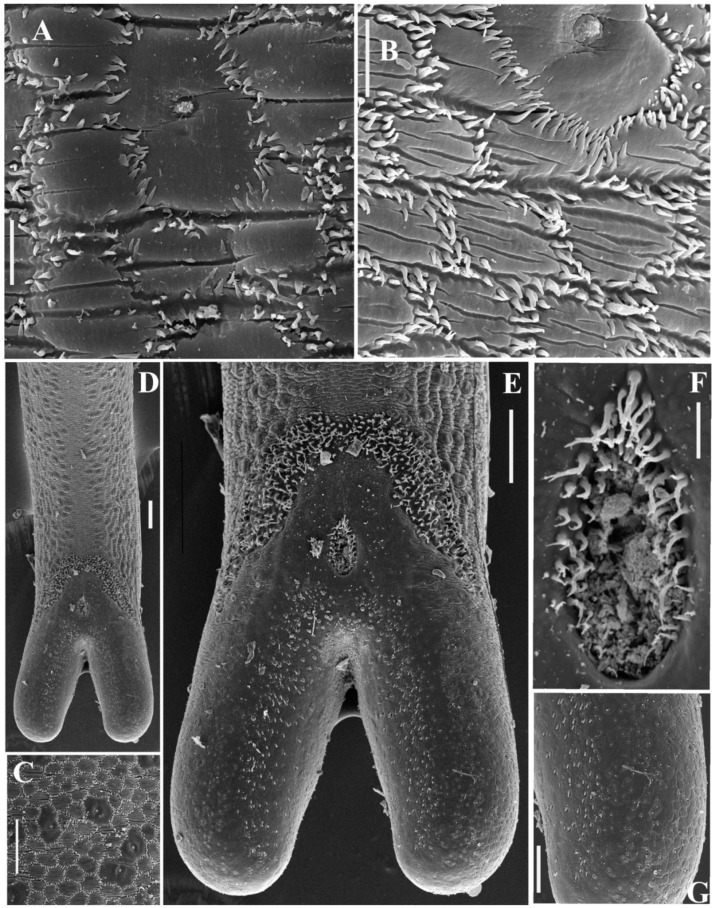
*Parachordodes pustulosus* cuticle surface, SEM (**A**—female, **B**–**G**—male). (**A**) two types of areoles, scale 10 μm; (**B**)—two types of areoles, scale 10 μm; (**C**)—two types of areoles under lower magnification, scale 50 μm; (**D**)—caudal lobes and ventral cuticle, scale 100 μm; (**E**)—cloacal area, ventrally, scale 100 μm; (**F**)—cloacal opening, scale 20 μm; (**G**)—postcloacal spines on the subventral part of lobe inner surface, scale 50 μm.

**Table 1 ijms-24-11411-t001:** *Parachordodes pustulosus* mitochondrial genome organization.

Gene	Strand	Gene Start	Gene Stop	Gene Length	Start Codon	Stop Codon	Inverted Repeat
*cox1*	+	10	1542	1533	ATG	TAA	no
*cox2*	+	1555	2232	678	ATG	TAA	no
*trnK*	+	2239	2300	62			no
*trnD*	+	2310	2363	54			no
*atp8*	+	2364	2483	120	ATC	TAA	no
*atp6*	+	2490	3173	684	ATG	TAA	yes
*cox3*	+	3176	3967	792	ATG	TAA	yes
*trnG*	+	3987	4044	58			no
*nad3*	+	4045	4380	336	ATA	T	no
*trnA*	+	4382	4436	55			no
*trnR*	+	4437	4487	51			no
*trnN*	+	4489	4546	58			no
*trnS1*	+	4548	4601	54			yes
*nad4l*	+	4602	4862	261	ATA	TAG	yes
*nad4*	+	5003	6184	1182	ATA	TAG	yes
*trnH*	+	6176	6237	62			no
*nad5*	+	6332	7660	1329	TTA	TAA	yes
*trnF*	+	7668	7722	55			yes
*trnW*	-	7719	7781	63			yes
*nad2*	-	7767	8738	972	TTA	TAA	yes
*trnM*	-	8754	8816	63			no
*trnQ*	+	8814	8872	59			no
*trnY*	+	8885	8939	55			no
*trnT*	+	8955	9010	56			no
*trnP*	-	9017	9083	67			yes
*nad6*	+	9160	9702	543	ATA	TAA	yes
*cytb*	+	9653	10,765	1113	ATA	TAG	yes
*trnS2*	+	10,765	10,821	57			yes
*trnE*	+	10,836	10,897	62			yes
*trnL1*	-	10,898	10,963	66			yes
*rrnL*	-	10,969	12,037	1069			yes
*trnV*	-	12,038	12,098	61			yes
*rrnS*	-	12,093	12,754	662			yes
*trnC*	-	12,755	12,810	56			no
*trnI*	-	12,822	12,871	50			no
*nad1*	-	12,873	13,778	906	TTG	TAG	yes
*trnL2*	-	13,782	13,847	66			no

**Table 2 ijms-24-11411-t002:** Nucleotide composition characteristics of the *Parachordodes pustulosus* mitochondrial genome.

	GC%	A%	T%	G%	C%
Whole genome	27	33	40	12	15
PCGs	29	32	39	12	17
tRNA	23	37	40	11	12
rRNA	21	38	41	9	12

## Data Availability

Mitochondrial DNA of *Parachordodes pustulosus* is available in GenBank (Accession Number OR003915). The raw data are deposited in GenBank in PRJNA989631 BioProject with SRA accession numbers SRR25099515 (RNA-seq) and SRR25099516 (DNA-seq).

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
