# Peer review of "Expression of Hairpin-Enriched Mitochondrial DNA in Two Hairworm Species (Nematomorpha)"

_ijms, 2023, doi:10.3390/ijms241411411_

Round 1

Reviewer 1 Report

This paper entitled “Expression of hairpin enriched mtDNA in hair worms” adds new knowledge to an existing knowledge base of Nematomorpha by updating and annotating the repeats in the mitochondrial genome of Parachordodes pustulosus. The following changes are recommended:

1.     Please add more detail to your title that encapsulates the contents of the article. Currently it reads as though it is a review paper. Expand mtDNA to mitochondrial DNA. Expand hair worms to the Nematomorpha or ideally the genus and species name since you are only tackling one species.

2.     Line 83: please change “proteins” to “protein coding genes”

3.     Lines 110 to 112: please state here the average in percentage even though it occurs in the table.

4.     Lines 151 to 154: it is not clear in line 151, if by genomic library you mean a transcriptomic library. In lines 152 and 153 you state that DNA-seq coverage is not affected by smaller inverted repeats and then clarify in 156 to 157 that the read depth of RNA seq libraries vary significantly. Please clarify the meaning of genomic library in line 151.

5.     Section 2.3.1 please use reduces or equivalent verb instead of “dips”.

6.     Line 201. please italicize the biological names

7.     line 250. change 13 to 3 in the heading of this section.

8.     Line 347: the results from spades - that is draft whole genome - is not presented in this paper

9.     Are the raw data available in GenBank? Please provide a reason for why raw data has not been deposited to the SRA database.

Overall the paper is well written.

Author Response

The authors acknowledge the positive attitude and valuable comments supplied by the Reviewer. We have tried to improve the text according to the Reviewer's comments.

  1. We removed the abbreviation from the title and specified the taxonomy of research object.
  2. Line 83: “proteins” was replaced by “protein coding genes”.
  3. Lines 110-112: the averages in percentage were stated.
  4. Line 151: the word “DNA” was added.
  5. The word “dips” was replaced by “drops” through the text.
  6. Line 201: Latin species names were italicized.
  7. Line 250: the number “13” was replaced by “3”.
  8. Line 347: We did not correct or improve the draft SPAdes assembly beyond the mitochondrial DNA contig. For this reason, we do not guarantee the draft nuclear assembly is free from different artifacts and do not publish it.
  9. The raw data are deposited in GenBank in PRJNA989631 BioProject with SRA accession numbers SRR25099515 and SRR25099516 (RNA-seq and DNA-seq, respectively).

Reviewer 2 Report

The paper addresses the scarcity of molecular data in the phylum Nematomorpha. The authors aim to shed light on the unique feature of long inverted repeats found in the mitochondrial genomes of nematomorphs, specifically focusing on Parachordodes pustulosus. The study explores the impact of these repeats on read coverage, provides phylogenetic analysis, and discusses potential discrepancies between traditional taxonomy and molecular data.

Strengths:

Comprehensive Investigation: The paper presents a thorough investigation of long inverted repeats in the mitochondrial genome of Parachordodes pustulosus. By utilizing both genomic and transcriptomic libraries, the authors provide valuable insights into the impact of these repeats on read coverage and highlight the limitations of short-fragment sequencing libraries.

Novel Data: The inclusion of Parachordodes pustulosus mitochondrial genome data adds to the limited existing molecular information on nematomorphs. This novel dataset contributes to the understanding of the genetic characteristics and evolutionary relationships within the phylum.

Phylogenetic Analysis: The paper's inclusion of phylogenetic inference using the newly obtained data is a significant strength. By comparing traditional taxonomic classification with molecular data, the authors identify discrepancies and propose changes that render several genera, including Parachordodes, as paraphyletic.

Clear Presentation: The paper is well-structured and effectively communicates its objectives, methods, and findings. The results are presented clearly, with appropriate figures and tables to support the data analysis.

Suggestions for Improvement:

·       Discussion of Functional Implications: While the paper successfully highlights the presence and impact of long inverted repeats, further exploration of the potential functional significance of these repeats would enhance the overall understanding of their role in the nematomorph life cycle.

·       Comparative Analysis: Including a comparative analysis of mitochondrial genomes from other nematomorph species, particularly those belonging to paraphyletic genera, would strengthen the discussion of the molecular data's implications for the traditional classification system.

·       Complete mitochondrial genome of Parachordodes pustulosus size is 14, 013 bp. Authors have not mentioned about the A+T-rich region or D-loop in the result and discussion part. A+T-rich is covered or not. Authors have to clearly mention in the text.

·       Total lengths of the genes are not given in the Table 1. Gene lengths can be included in the separate column in the table. A+T-rich region or D-loop are not included in the table.

·       Authors have mentioned in line no 358. “Mitochondrial genome map was constructed using OGDRAW software. However, the mitochondrial circular form was not given in the text.

·       The DNA isolation methods and techniques were not included in the material and methods.  Separate paragraph must be included. 

·       In my observation gene order and strand, the tRNA genes trnF, trnY, trnI, trnL2 are inverted translocated. The gene nad5 inverted. The genes trnT, trnE, trnC and nad1 are translocated. The lines from 115-121 can be rewritten based on the inverted translocation of the genes. 

·       Future Research Directions: Providing suggestions for future research directions or potential experiments could help guide further investigations into nematomorph biology, genomics, and evolutionary relationships.

Author Response

The authors acknowledge the positive attitude and valuable comments and suggestions supplied by the Review Report. We present a revised version of the manuscript as directed by the Reviewer and detail the changes hereby.

  1. Thank you for this suggestion! We are already thinking about possible approaches for detection of functional significance of inverted repeats. We suppose that direct RNA sequencing without cDNA stage (especially with long reads) can shed light on real nematomorph mitochondrial transcripts. In that case we will be able to predict the influence of hairpins on mtDNA expression in nematomorphs more clearly.
  2. We added the comparative analysis of polyadenylation in two nematomorphs and the comparative analysis of putative control region structures in six available mitochondrial genomes of Nematomopha.
  3. We discussed a control region position in nematomorphs and added a new figure with a putative control region in mitochondrial genome of P. pustulosus. Unfortunately, we do not have experimental data, which could verify the D-loop localization.
  4. Gene lengths were included in the separate column in the Table 1.
  5. Mitochondrial circular map was added to manuscript; wrongly mentioned program OGDRAW was replaced by Circos.
  6. The DNA isolation methods and techniques were described in more details.
  7. We retained Table 1 containing the coding strand information for each gene and removed the listing of tRNA genes from the text.

Reviewer 3 Report

The authors obtained and annotated the repeats in the mt genome of Parachordodes pustulosus, and investigated the impact of inverted repeats on the read coverage of the mitogenome. The paper is well written and the results are clearly presented, and I support the publication of this paper. I only have several minor points.

1. Line 62, the comma should be replaced by a full stop.

2. Line 89, what does the character after ”trnP“ mean?

3. Line 201, Gordionus alpestris and Parachordodes pustulosus should be italicized.

4. Lines 423-427, could the authors describe how the model for phylogenetic construction was selected? What softwares were used?

Author Response

The authors are grateful for the positive attitude and valuable comments of the Reviewer. We have corrected the text according to all the instructions of the Reviewer.

  1. Line 62: the comma was replaced by a full stop.
  2. Line 89: wrong character was removed from the text.
  3. Line 201: Latin species names were italicized.
  4. Lines 423-427: the method of selection of evolutionary models for phylogenetic inferences was described.